# RETRO-TAS, a Retrospective Observational Study of Trifluridine/Tipiracil in Chemorefractory Metastatic Colorectal Cancer

**DOI:** 10.3390/biomedicines11051267

**Published:** 2023-04-24

**Authors:** Anna Koumarianou, Anastasios Ntavatzikos, David Symeonidis, Christos Vallilas, Maria Giannakakou, Georgios Papaxoinis, Spyridon Xynogalos, Ioannis Boukovinas, Stamatina Demiri, Katerina Kampoli, Georgios Oikonomopoulos, Epaminontas Samantas, Eleni Res, Nikolaos Androulakis, Georgia Vourli, Ioannis Souglakos, Michalis Karamouzis

**Affiliations:** 1Hematology Oncology Unit, Fourth Department of Internal Medicine, Attikon University Hospital, National and Kapodistrian University of Athens, 12461 Athens, Greece; dmaal2@yahoo.gr (A.N.); katerinakamboli@yahoo.gr (K.K.); 2Medical Oncology, “Metaxas” Cancer Hospital, 18537 Piraeus, Greece; davidsymeonidis@yahoo.com (D.S.); sxyn@otenet.gr (S.X.); 3School of Medicine, National and Kapodistrian University of Athens, 11527 Athens, Greece; 4Medical Oncology, “Agii Anargyri” Cancer Hospital, 14564 Athens, Greecenellieres@yahoo.gr (E.R.); 5Medical Oncology, “Agios Savvas” Cancer Hospital, 11522 Athens, Greece; georgexoinis@gmail.com (G.P.); matina.demiri@gmail.com (S.D.); 6Oncology Department, Bioclinic of Thessaloniki, 54622 Thessaloniki, Greece; ibouk@otenet.gr; 7Medical Oncology, “METROPOLITAN” Hospital, 18547 Piraeus, Greece; 8Medical Oncology Unit, Pananio-Venizelio General Hospital of Heraklion, 71409 Heraklion, Greece; 9Department of Hygiene, Epidemiology and Medical Statistics, Medical School, National and Kapodistrian University of Athens, 11527 Athens, Greece; gvourli@med.uoa.gr; 10Department of Medical Oncology, University Hospital of Heraklion, 71013 Crete, Greece; johnsougl@gmail.com

**Keywords:** colorectal cancer, metastasis, KRAS, NRAS, BRAF, HER2, MSI, chemotherapy

## Abstract

Background: Trifluridine/tipiracil (FTD/TPI) is an oral antimetabolite agent comprised of trifluridine, a thymidine-based nucleoside analogue that inhibits cell proliferation following its incorporation into DNA, and tipiracil that helps maintain the blood concentration of trifluridine by inhibiting the enzyme thymidine phosphorylase which inactivates trifluridine. It is approved as a third-line treatment option for patients with metastatic colorectal cancer (mCRC) and is administered at 35 mg/m^2^ two times daily from day 1 to 5 and from day 8 to 12 every 28 days. The aim of this investigator-initiated retrospective study (RETRO-TAS; NCT04965870) was to document real-world data on the clinical efficacy of FTD/TPI in patients with chemorefractory mCRC. Methods: The clinical characteristics of patients with mCRC treated with FTD/TPI in 8 Cancer Centres were collected to assess physician’s choice in the third or beyond line of treatment as well as the duration of treatment, dose modification, and toxicity. In addition, other important prognostic features related to mCRC such as molecular profile, performance status (PS), and primary site were analyzed. Statistical analysis for progression-free survival (PFS), overall survival (OS), 6-/8-month PFS rate and disease control rate (DCR) along with Cox regression model, Kaplan–Meier curves, and log-rank tests were carried out by using Stata/MP 16.0 for Windows. Results: From October 2018 to October 2021, a total of 200 patients with mCRC and a median age of 67.0 (IQR 58.0, 75.0) years were treated with FTD/TPI. Τhe median follow-up time was 14 months (IQR 7, 23), 158 PDs and 106 deaths were reported at the time of this analysis. Of all the patients, 58% were males and 58% had mCRC at diagnosis. The molecular analysis identified mutations in KRAS (52%), NRAS (5%), HER2 (3.5%), BRAF (3.5%), and MSI (9%). Previous treatments included radical surgery in 51.5% and adjuvant chemotherapy in 39.5% of patients. FTD/TPI was administered in the third- (70.5%), fourth- (17.0%), or fifth-line (12.5%) treatment setting. Serious adverse events related to FTD/TPI included neutropenia (2%), anaemia (1%), thrombocytopenia (0.5%), diarrhoea (0.5%), nausea (0.5%), and fatigue (4%). A reduction of FTD/TPI dose, delay of next cycle initiation, and shorter duration were reported in 25%, 31%, and 14.5% of patients, respectively. Of all the patients 71.5% received FTD/TPI as monotherapy, 24.5% in combination with bevacizumab, and 4.0% with an anti-EGFR agent. The median FTD/TPI treatment duration was 119.5 days and 81% of patients discontinued treatment due to progressive disease. The DCR recorded by investigators’ assessment was 45.5%. The median PFS was 4.8 and the median OS was 11.4 months. The 6- and the 8-month PFS rate was 41.4% and 31.5%, respectively. In the multivariate analysis, PS > 1 and presence of liver and lung metastasis were adversely associated with PFS and OS whereas mutational status and tumor sidedness were not. Conclusions: RETRO-TAS is a real-world observational study that confirms and adds on the findings of the pivotal RECOURSE Phase III study in relation to the efficacy of FTD/TPI in the third-line setting and in all subgroups of patients regardless of mutational status and sidedness.

## 1. Introduction

Colorectal cancer (CRC) is the third most common cancer with a global incidence of 1.85 million new cases every year [1]. Approximately 50% of CRC patients will present with distant metastasis (mCRC) and of these less than 20% will have a 5-year overall survival (OS).

During recent years, as we have witnessed significant improvements in targeted therapies and localized approaches targeting the metastatic burden, an increasing number of patients with mCRC can receive three or more lines of therapies. Many different parameters, including performance status, age, tumor sidedness, optimal staging, and genomic profiling, can be used to guide treatment selection while maintaining quality of life in the context of multidisciplinary tumor boards. Although cures remain uncommon, more patients can derive benefits and fewer are exposed to toxicity from ineffective therapies.

The recognition of the genomic profiling of a patient, such as RAS, BRAF, and microsatellite instability, is crucial for treatment stratification. Based also upon the available current guidelines, most of these patients will receive a first and a second-line therapy based on fluoropyrimidines, a thymidylate synthase inhibitor, and oxaliplatin (CAPOX/FOLFOX) or irinotecan (FOLFIRI/CAPIRI) combined with vascular endothelial growth factor or epidermal growth factor inhibitors [1,2]. Even though the genomic profiling facilitates the selection of the first- and second-line therapies, the selection of further lines of treatment can become less clear.

Trifluridine/tipiracil (FTD/TPI) or TAS-102 is an orally administered compound of a thymidine-based nucleoside analogue (trifluridine) and a thymidine phosphorylase inhibitor (tipiracil) that prevents the rapid degradation of trifluridine, thus allowing the maintenance of adequate plasma concentrations.

The pivotal Phase III study RECOURSE investigated the oral agent FTD/TPI in the third and beyond line of treatment indicating significant clinical activity in heavily pre-treated mCRC patients, including those whose disease was refractory to fluorouracil, and was associated with few serious adverse events mostly related to myelosuppression [3]. The RECOURSE study showed that compared with best supportive care (BSC), FTD/TPI increased the median overall survival (mOS) from 5.3 to 7.1 months (*p* < 0.001) and median progression-free survival (mPFS) from 1.7 to 2.0 months (*p* < 0.001).

Based on a Phase II study by Yoshino et al., FTD/TPI was initially approved for refractory mCRC in Japan in March 2014 [4]. Following the results of the pivotal RECOURSE Phase III study, FTD/TPI was additionally approved in the United States (September 2015) and Europe (April 2016) [3,5,6]. A more recent Phase III trial, the TERRA study, carried out in an entirely Asian patient population, confirmed the earlier reported positive results [7]. In all of these three studies, patients were previously treated with two or more chemotherapy regimens including oxaliplatin, fluoropyrimidine, and irinotecan and had received FTD/TPI as a third or more advanced line of treatment. A difference between RECOURSE and the Yoshino or the TERRA studies was that patients with KRAS mutation or KRAS wild-type must also have had bevacizumab or an anti-EGFR antibody, respectively, prior to FTD/TPI therapy. More recently, a single arm Phase I–II study combining FTD/TPI with the antiangiogenic agent bevacizumab indicated encouraging responses [8]. There were two additional Phase II randomized studies comparing the use of trifluridine/tipiracil and bevacizumab to single agent FTD/TPI in patients with mCRC that indicated significant and clinically relevant improvement in progression-free survival with tolerable toxicity [9,10,11].

The primary objectives of previous Phase II and III trials, such as RECOURSE were to explore safety and efficacy of FTD/TPI in groups of patients who satisfied the selected inclusion and exclusion criteria. In the clinical practice, however, antineoplastic drugs are typically used in a less selected manner to a rather heterogeneous group of patients [12].

Consequently, the effectiveness of FTD/TPI may differ in the real-world setting when compared to the results from the clinical trials [13]. To confirm the efficacy of FTD/TPI, it is necessary to assess the results from studies that describe common clinical practice with unselected patients. A recent meta-analysis of real-world data from 64 Japanese and European centres indicated that the effectiveness of FTD/TPI monotherapy in late-stage mCRC is consistent with the efficacy outcomes of the RECOURSE study but the survival is inferior to the outcomes of the two Asian studies by Yoshino et al. and the TERRA trial [14]. This difference points out to a possibly different molecular profile of the disease between Asian and Caucasian populations and underlines the need for further evaluation from single country registries.

FTD/TPI administration was approved by the European Medicine Agents as a monotherapy for patients with mCRC who either are not eligible for 5FU-, oxaliplatin-, or irinotecan-based treatments or have experienced disease progression to these chemotherapies, anti-VEGF, and anti-EGFR agents.

The aim of this study was to record clinical practice and to collect real-world data on the clinical efficacy of FTD/TPI in Greek patients with mCRC.

## 2. Materials and Methods

### 2.1. Study Design

This was an investigator-initiated, observational, retrospective study, designed to record clinical practice and to collect real-world data on the clinical efficacy of FTD/TPI treatment in the third-line and beyond and in chemo-resistant metastatic colorectal cancer. This retrospective study was conducted in accordance with Good Clinical Practice, as defined by the International Conference on Harmonisation, Good Pharmacoepidemiological Practice, and in accordance with the ethical principles underlying European Union Directive 2001/20/EC. The study was conducted in compliance with the prespecified protocol and received Institutional Review Board/Independent Ethics Committee (IRB/IEC 14/8-9-2020) approval opinion prior to initiation. This study (RETRO-TAS) was registered in clinicalTrials.gov (NCT04965870) [15]. Part of the results of the RETRO-TAS study have been presented as a poster at ESMO-GI 2022 [16].

### 2.2. Patient Population

The study aimed to collect data from medical files of patients age ≥18 years old, with available data on previous chemotherapy lines based on histologically confirmed mCRC. There were no additional exclusion criteria. The number of participating centres was eight and the planned number of included patients was 200. The time period of patient timeline inclusion in the study was June 2021–December 2021.

### 2.3. Study Treatments

Patients were treated with oral FTD/TPI 35 mg/m^2^ twice daily (after morning and evening meals) on days 1–5 and 8–12 of each 28-day cycle. The starting dose of 35 mg/m^2^ was maintained throughout the treatment period as long as the patient was receiving benefit from FTD/TPI and no adverse events leading to dose reduction occurred. Dose adjustments and dose delays were based on clinical criteria such as individual safety and tolerability and the recommendations included in the product information. Patients continued receiving treatment until one or more of the following criteria for treatment discontinuation were met: patient/physician decision, disease progression, death due to disease progression or unacceptable toxicity.

### 2.4. Study Endpoints

The primary endpoint of the study was PFS. Secondary endpoints included the OS, PFS rate at 6 months and 8 months, duration of treatment with FTD/TPI, disease control rate (DCR), and efficacy endpoint: PFS and OS in relation to (i) the mutational status and (ii) the number of metastases and time from metastatic progression in three scenarios as follows: (a) ≤2 metastatic sites, low metastatic burden, and >18 months from first diagnosis of metastasis; (b) ≤2 metastatic sites, low metastatic burden and <18 months from first metastasis but no liver metastasis; (c) >3 metastatic sites with high mutational burden and <18 months from first metastasis. Additional endpoints included (i) the registry of adverse events related to treatment, evaluation of the onset in each treatment cycle, intensity, and seriousness and (ii) the evaluation of the proportion of patients who had a modification of the dose or discontinued the treatment due to adverse events.

### 2.5. Study Safety Assessments

The safety evaluations were focused on adverse events (AEs) and laboratory assessments. Adverse events were coded according to the Medical Dictionary for Regulatory Activities (MedDRA) terminology and the severity of the toxicities were graded according to the Common Terminology Criteria for Adverse Events (CTCAE) criteria, v5.0 where applicable. All AEs were summarized (incidence) and listed by the System Organ Class (SOC), toxicity/severity grade, and causal relationship to study medication. In addition, separate summaries of SAEs and Grade 3 and 4 AEs were included. Haematological and chemistry laboratory parameters were graded according to the NCI CTCAE v.4.03 criteria, where applicable. In addition, worst severity grade, time to event, and time to resolution were summarized.

### 2.6. Statistical Analysis

Continuous variables were summarized as medians and interquartile ranges (IQRs) while counts and corresponding percentages were calculated for categorical variables. Alive patients and those lost to follow-up were censored on the last date on which they were known to be alive. Patients alive at the end of the study were censored at this timepoint. All statistical comparisons were performed using non-parametric tests: Fisher’s exact tests in case of frequencies’ comparisons, Mann–Whitney, and Kruskal–Wallis tests for the comparison of median values between two groups and more than two groups, respectively. Survival times were estimated by the Kaplan–Meier method. Risk factors for PFS and OS were evaluated with Cox regression models. The statistical software applied was STATA/MP 16.0 for Windows (StataCorp LLC, College Station, TX, USA).

#### Study Outcomes

Progression-Free Survival (PFS): PFS is defined as the time interval from initiation FTD/TPI to the first date of documented tumor progression or death from any cause, whichever occurs first, before the initiation of a new anti-cancer therapy.

Imaging Responses and Disease Control Rate (DCR): Imaging confirmation of response by computerized tomography of abdomen and pelvis according to local assessment was carried out using RECIST 1.1 criteria. In the case of SD, measurements must have met the SD criteria at least once after treatment start at a minimum interval of 6 weeks. To be assigned a status of PR or CR, changes in tumor measurements must be confirmed by repeat assessments that should be performed no less than 4 weeks after the criteria for response are first met. The DCR is defined as the proportion of patients with objective evidence of complete response (CR), the proportion of patients with objective evidence of partial response (PR), and the proportion of patients with objective evidence of stable disease (SD).

Overall Survival: Overall survival is defined as the time interval from initiation of treatment to the date of death due to any cause.

Exploratory analyses: Univariate and multivariate (if appropriate) Cox regression analyses were also be performed to explore the potential prognostic factors among basic clinicopathological characteristics such as performance status (PS) according to Eastern Cooperative Oncology Group, tumor sideness, presence of metastases at diagnosis, time from diagnosis, previous treatments, sites of metastases and molecular profiling (HER2, KRAS, NRAS, MSI, BRAF), and line of therapy with FTD/TPI with respect to OS and PFS given that the final sample size allows for such analyses. Time-to-event distributions were estimated by the KaplanMeier method and comparisons between groups were assessed by the two-sided log-rank test.

PFS rate at 6 and 8 months (%): The primary efficacy endpoint is the progression-free survival (PFS) rate at 6 and 8 months corresponding to the percentage of patients surviving without any documented progression of the disease at 6 and 8 months after treatment initiation.

Determination of best overall response: Best overall response is the best confirmed response recorded from initiation of treatment until documented disease progression and before the initiation of a new anti-cancer therapy. For each patient, organ involvement, best overall response, and date of progression were assessed.

## 3. Results

### 3.1. Baseline Characteristics

From October 2018 to October 2021, 200 patients with a median age at diagnosis of 63.7 years (IQR 54.2, 72.1) and at FTD/TPI treatment initiation was 67.0 (IQR 58.0, 75.0). At the time of the analysis, the median follow-up time was 14 months (IQR 7, 23), and 158 PDs and 106 deaths due to mCRC were recorded. Patient clinicopathologic characteristics on FTD/TPI initiation are shown in Table 1. Details are provided on the PS, American Society of Anesthesiologists (ASA) classification score, molecular profiling, sideness of CRC, presence of metastatic disease at diagnosis, time of FTD/TPI initiation from CRC diagnosis, sites of metastases on FTD/TPI initiation, and line of FTD/TPI treatment. The most frequent primary site was the left colon (45%) followed by the rectum (36.5%). KRAS mutation was detected in 52% and MSI in 9% of patients. None of the patients included in the study had a positive family history for Lynch or any other hereditary syndrome. A total of 58% of patients had metastatic disease at the time of diagnosis. Radical surgery and adjuvant chemotherapy were delivered in 51.5% and 39.5% of patients, respectively. In total, 86% of patients had metastases in three or more different organ sites, most commonly the liver, lung, and distant lymph nodes at the time of FTD/TPI treatment.

### 3.2. Study Treatment

Details on FTD/TPI treatment are shown in Table 2. FTD/TPI was administered according to the approved doses as a third- (70.5%), fourth- (17.0%), or fifth-line (12.5%) of therapy. FTD/TPI was administered as monotherapy (in 71.5% of patients), in combination with bevacizumab (24.5%), or with an anti-EGFR agent (4.0%). The time of FTD/TPI administration was less than 18 months from CRC disease diagnosis in 30% of patients. For patients treated with FTD/TPI in the third-line setting, previous lines of treatment most frequently included 5-FU or capecitabine (100%) combined with oxaliplatin (84.5%) and bevacizumab (94%). The median duration of FTD/TPI therapy was 119.5 days (4 months; IQR 2.85-7.13) and at the time of the analysis 158 patients (79%) had discontinued therapy, most commonly (81%) due to progressive disease. With respect to treatment schedule, 25%, 31%, and 14.5% of patients experienced a dose reduction, a delay of cycle, and a shorter than 10 days therapy, respectively, due to toxicity.

### 3.3. Efficacy

Based on best objective responses to FTD/TPI therapy according to locally assessed imaging by RECIST criteria, the disease control rate (DCR) was 45.5%.

With respect to FTD/TPI treatment, the median PFS time recorded was 4.8 and the median OS was 11.4 months (Figure 1). The 6- and the 8-month PFS rates were found to be 41.4% and 31.5%, respectively.

Table 3 indicates the characteristics of significance in relation to survival analysis. The performance status (PS) >1 and presence of metastatic disease in both the liver and lung were found to be adversely associated with both PFS and OS. The presence of metastatic disease at diagnosis was negatively associated with OS.

Table 4 indicates the associations that were shown to be significant or of interest in the univariate analysis but were not confirmed in the multivariate model. Patients receiving FTD/TPI in the third-line had a significant trend for improved PFS and OS compared to those receiving it in more advanced lines. The rectum as the primary site of mCRC had statistically significant better PFS compared to the right colon primary and a better OS although this did not reach statistical significance. Combination treatment of FTD/TPI with bevacizumab had an improved PFS and OS, although this did not reach statistical significance compared to FTD/TPI monotherapy.

The investigation of potential associations of other pathologic features such as the presence of mucus >50%, the mutational status for KRAS, NRAS, BRAF, HER2, and MSI were not found to associate with objective responses, PFS or OS.

### 3.4. Safety

Adverse events of interest are shown in Table 5. The most common adverse events included myelotoxicity (neutropenia 40.5%, anemia 36%), fatigue (36%), and nausea (24%). Serious adverse events reported were neutropenia (2%), anaemia (1%), thrombocytopenia (0.5%), diarrhoea (0.5%), nausea (0.5%), and fatigue (4%). Treatment discontinuation due to toxicity was reported only in four patients (2.5%) that were included in the study (Table 2).

## 4. Discussion

The RETRO-TAS study provides real-world population-based data on FTD/TPI treatment in patients with mCRC. The median OS and PFS from initiation of FTD/TPI in the total cohort were 11.4 and 4.8 months, respectively. The RETRO-TAS study is a retrospective study and the first to demonstrate such a significant survival benefit in the third-line and beyond therapeutic setting. Although this finding may be multifactorial, in our opinion the most important component contributing towards these exceptional survival benefits is the line of treatment. In fact, 70.5% of patients received FTD/TPI in the third-line whereas 17.5% received it in the fourth-line and only 12% in the fifth-line of therapy.

FTD/TPI has been approved for the third-line and beyond treatment of mCRC patients who have previously received combination chemotherapy with fluoropyrimidine, oxaliplatin, irinotecan, and targeted therapies with antiangiogenics or an anti-EGFR in KRAS wild-type subtypes [17]. The basis of approval was the RECOURSE randomized Phase III study that showed an OS of 7.1 months in the FTD/TPI arm versus 5.3 months in the BSC arm [3]. The PFS benefit in the same study was 2.0 months for patients in the FTD/TPI arm and 1.7 months for those in the BSC arm. By looking at the details of patient characteristics of this pivotal study, only 18% of patients received FTD/TPI in the third-line whereas 22% and 60% received it in the fourth- and in the fifth-line of treatment, respectively [3].

A previous randomised Phase II study from Japan including a total of 169 patients indicated a slightly better OS at 9 months [4]. In this study, 85% of patients allocated in the FTD/TPI arm belonged in the fourth-line and beyond.

Another Phase III study from China, (the TERRA trial) including 406 patients showed a statistically significant OS and PFS of FTD/TPI over placebo of 7.8 and 2 months, respectively [7]. In this study, only 23% of patients had received FTD/TPI in the third-line setting whereas 27% and 50% had received it in the fourth and beyond lines of therapy.

Similarly, a meta-analysis of a published series of metastatic colorectal cancer patients treated with FTD/TPI showed a pooled median OS and PFS of 6.6 months (95% confidence interval: 6.1–7.1 months) and 2.2 months (95% confidence interval: 2.1–2.3 months), respectively [14]. However, these findings have not been related to the clinicopathological characteristics of included patients so safe conclusions in relation to the line setting of FTD/TPI treatment cannot be made.

PRECONNECT is a Phase IIIb study including 793 patients, of whom 36.1% received third-line FTD/TPI, that has indicated a median PFS at 2.8 months [18]. The study has no mature data reporting the median OS.

The ROS real-world study included 379 patients and reported a median PFS and OS of 3 and 8 months, respectively. In the same study, 66.8% of patients were previously treated with ≥ three lines of treatment indicating a heavily pre-treated patient population [19].

Previous studies investigating prognostic risk models in patients with mCRC treated with FTD/TPI have concluded that metastatic disease at diagnosis, PS ≥ 2 and liver metastases are negatively associated with survival in mCRC [3,4,7,18,19,20,21,22,23]. Clinical trials do not include patients with PS ≥ 2 as a worse PS would be expected to have a negative impact on outcomes since on its own it has been shown to be a prognostic factor for patients with advanced cancer [24]. However, real-world clinical studies commonly include patients with PS ≥ 2 and a previous study has clearly indicated that patients with >2 performance status still benefit from treatment with FTD/TPI [21]. The multivariate analysis of our study, having adjusted for potential confounders, shows that PS 0–1 is an independent factor that reduces the risk of disease progression and death in patients who received FTD/TPI. All these data together indicate that FTD/TPI may be beneficial in patients with mCRC and PS > 2, although it would be advisable to administer this drug within its approved indication in the third-line and when patients are expected to have a better PS. Other important risk factors identified in the multivariate analysis of our study and were associated negatively with PFS and OS is the presence of metastatic sites in the liver and lung and the presence of metastases at CRC diagnosis. The presence of liver metastasis, high tumor burden, and metastatic disease has been previously negatively associated with survival both in a clinical trial and a real-world study of FTD/TPI treatment [19,25].

In our study, the DCR was 45.5% and this was similar to the DCR of 44% that was recorded in both the RECOURSE Phase III study and the JapicCTI-090880 Phase II study, confirming the efficacy of FTD/TPI to induce objective responses in the third-line setting [3,4]. Two additional important aspects that were indicated in the univariate analysis of our study is that combination of FTD/TPI with bevacizumab and rectal primary could be associated with enhanced PFS and OS. Rectal primary has not been identified as a potential biomarker in previous studies [20] but combination with bevacizumab has been indicated as an important adjunct to FTD/TPI in a meta-analysis, in a Phase II, and the SUNLIGHT Phase III clinical trial [11,26,27].

Our study indicated an acceptable toxicity profile, including mostly fatigue, hematologic, and gastrointestinal adverse events, that concurs with that reported by previous phase II/III and real-world studies [3,4,7,18,22]. The most common severe toxicity was fatigue (eight patients) and neutropenia (four patients), all manageable according to standard common practice. The proportion of patients in our study that discontinued treatment due to toxicity was low at 2.5% (four patients), but dose adjustments were performed including dose reduction (25% of patients), delays of next cycle initiation (31%), and shorted than 10 days treatment duration (14.5%) based on patients/physicians preferences and prescribing information in agreement with previously reported data [17].

Real-world studies can reflect more clearly routine clinical practice than Phase III trials. However, the small number of patient subgroups and the retrospective data extraction from medical charts entailed limitations due to data availability. Furthermore, in our study, the evaluation of clinical responses was carried out by each participating centre and not centrally and this may have led to evaluation bias in the objective response rates. This may particularly have influenced the correct assessment of patients achieving some response but not clearly reaching the definition of partial response. The advantage of real-world studies such as RETRO TAS is that the administration of drugs such as FTD/TPI is examined in broader patient populations, with poorer performance status or comorbidities, and prescribers have fewer restrictions to modify doses, regimens, or concomitant therapy. This can be translated as real-world studies have the potential of higher external validity which, however, is obtained at the expense of the internal validity [14].

## 5. Conclusions

FTD/TPI has a well-established efficacy with an easily manageable safety profile in patients with mCRC who are refractory or intolerant to first- and second-line strategies including chemotherapeutics such as fluoropyrimidines, irinotecan, and oxaliplatin with targeted agents. The results of our study are consistent with the results of the RECOURSE study and potentiate the role of FTD/TPI as an important treatment in all subgroups of mCRC pre-treated patients irrespective of primary site, histologic, and molecular characteristics, particularly in the third-line setting where the effect seems to be more beneficial. Combination regimens may further improve the therapeutic yield of FTD/TPI and several clinical trials are underway.

## Figures and Tables

**Figure 1 biomedicines-11-01267-f001:**
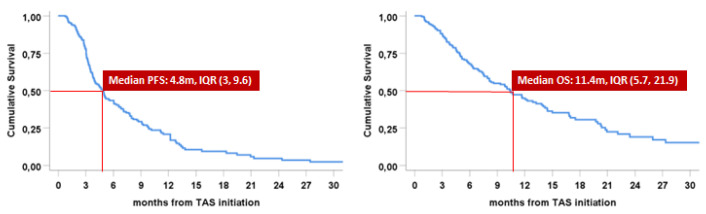
Progression-free survival (PFS) and overall survival (OS).

**Table 1 biomedicines-11-01267-t001:** Patient and disease characteristics on trifluridine/tipiracil initiation in mCRC.

Parameters	Number of Patients N (%)
Total number of patients	200 (100%)
Median age	67 years
Age >65 years	115 (57.5)
Female Gender	84 (42.0)
Metastatic at diagnosis	116 (58.0)
Performance Status (ECOG)	
0 or 1	135 (67.5)
2 or 3	65 (32.5)
ASA Score	
Normal healthy patient	109 (54.5)
Mild systemic disease	78 (39.0)
Severe systemic disease	12 (6.0)
Constant threat to life	1 (0.5)
Tumor Site	
Right	37 (18.5)
Left	90 (45.0)
Rectum	73 (36.5)
Presence of Mucus >50%	25 (12.5)
Molecular Markers	
Her2-Positive	7 (3.5)
Microsatellite Instability	18 (9.0)
KRAS Mutant	104 (52.0)
NRAS Mutant	10 (5.0)
BRAF Mutant	7 (3.5)
Patients with	
No molecular marker	73 (36.5)
Any marker detected	127 (63.5)
Metastasis on treatment initiation	
Liver Metastatic Sites	
No	46 (23.0)
Yes	154 (77.0)
Lung Metastatic Sites	
No	95 (47.5)
Yes	105 (52.5)
Bone Metastatic Sites	
No	172 (86.0)
Yes	28 (14.0)
Other Metastatic Site	
No	141 (70.5)
Yes	59 (29.5)
Metastatic sites	
1 or 2	28 (14.0)
3 or more	172 (86.0)

**Table 2 biomedicines-11-01267-t002:** Details of trifluridine/tipiracil (FTD/TPI) treatment in mCRC.

FTD/TPI Scheme	Number of Patients N (%)
Total number of patients	200 (100%)
Monotherapy	143 (71.5)
FTD/TPI + bevacizumab	49 (24.5)
FTD/TPI + anti-EGFR	8 (4.0)
*Line of FTD/TPI*	
3rd	141 (70.5)
4th	34 (17.0)
5th	25 (12.5)
Previous lines of therapy when 3rd-line FTD/TPI	
5FU or capecitabine	200 (100)
Oxaliplatin	169 (84.5)
Irinotecan	137 (68.5)
Bevacizumab	188 (94)
Anti-EGFR	78 (39)
Aflibercept	39 (19.5)
FTD/TPI initiation	
>18 months from diagnosis	140 (70.0)
<18 months from diagnosis	60 (30.0)
Dose Reduction	
No	150 (75.0)
Yes	50 (25.0)
*Delay of Cycle*	
No	138 (69.0)
Yes	62 (31.0)
Shorter Treatment Duration (Less than 10 days)	
No	171 (85.5)
Yes	29 (14.5)
FTD/TPI Discontinuation	
No	42 (21.0)
Yes	158 (79.0)
Reason of FTD/TPI discontinuation	
Patient decision	3 (1.9)
Doctor’s decision	2 (1.3)
Toxicity	4 (2.5)
PD	128 (81.0)
Death	21 (13.3)
Duration of FTD/TPI treatment [median days (IQR)]	119.5 (85.5, 214.0)

**Table 3 biomedicines-11-01267-t003:** Multivariate Cox regression for progression-free survival and overall survival of trifluridine/tipiracil treatment in mCRC.

Risk Factor	HR 95% C.I.	*p*-Value
Progression-Free Survival		
Performance Status (ECOG)		
0 or 1 *	1	
2 or 3	2.686 (1.893, 3.813)	<0.001
Metastatic Sites		
Other *	1	
Both Liver and Lung Metastases	1.536 (1.108, 2.128)	0.010
Overall Survival		
Performance Status (ECOG)		
0 or 1 *	1	
2 or 3	2.648 (1.765, 3.972)	<0.001
Metastatic Sites		
Other *	1	
Both Liver and Lung Metastases	1.501 (1.010, 2.229)	0.044
Metastatic at diagnosis		
No *	1	
Yes	1.561 (1.049, 2.322)	0.028

* Reference Category.

**Table 4 biomedicines-11-01267-t004:** Univariate Cox regression for progression-free survival and overall survival of trifluridine/tipiracil (FTD/TPI) treatment in mCRC (risk factors not retained in the multivariate Cox regression model).

Risk Factor	HR 95% C.I.	*p*-Value
Progression-Free Survival		
FTD/TPI monotherapy *	1	
FTD/TPI + Bevacizumab	0.761 (0.521, 1.112)	0.158
FTD/TPI + anti-EGFR	0.784 (0.364, 1.689)	0.535
3rd line FTD/TPI *	1	
4th line FTD/TPI	1.185 (0.770, 1.824)	0.439
5th line FTD/TPI	2.484 (1.551, 3.980)	<0.001
Right Colon *	1	
Left Colon	0.774 (0.507, 1.181)	0.234
Rectum	0.632 (0.403, 0.991)	0.046
Overall Survival		
ASA Score Normal or Mild *	1	
ASA score Severe	0.459 (0.187, 1.130)	0.090
FTD/TPI monotherapy *	1	
FTD/TPI with Bevacizumab	0.724 (0.456, 1.147)	0.169
FTD/TPI + anti-EGFR	0.653 (0.261, 1.634)	0.362
3rd line FTD/TPI *	1	
4th line FTD/TPI	1.057 (0.622, 1.797)	0.838
5th line FTD/TPI	1.870 (1.097, 3.188)	0.021
Right Colon *	1	
Left Colon	0.851 (0.511, 1.417)	0.535
Rectum	0.615 (0.354, 1.068)	0.084

* Reference Category.

**Table 5 biomedicines-11-01267-t005:** The number of patients presenting toxicity, according to MedDRA, recorded on trifluridine/tipiracil treatment in the entire cohort of 200 patients. Grade 4 toxicity was not observed.

	Grade 1/2 N (%)	Grade 3 N (%)	Total N (%)
**Anaemia**	70 (35.0)	2 (1.0)	72 (36.0)
**Neutropenia**	77 (38.5)	4 (2.0)	81 (40.5)
**Thrombocytopenia**	26 (13.0)	1 (0.5)	27 (13.5)
**Fatigue**	64 (32.0)	8 (4.0)	72 (36.0)
**Nausea**	47 (23.5)	1 (0.5)	48 (24.0)
**Diarrhoea**	46 (23.0)	1 (0.5)	47 (23.5)
**Vomiting**	40 (20.0)	-	40 (20.0)
**Neuropathy**	17 (8.5)	-	17 (8.5)
**Hand and Foot Syndrome**	11 (5.5)	-	11 (5.5)
**Skin Rash**	8 (4.0)	-	8 (4.0)
**Mucositis**	8 (4.0)	-	8 (4.0)

N: number of patients presenting toxicity.

## Data Availability

Data are available through the participating hospital registries and the Hellenic Study Group of Psychoneuroimmunology in Cancer following appropriate approval. Derived data supporting the findings of this study are available from the corresponding author A.K. on request after appropriate ethical approval.

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
