# Peer review of "RETRO-TAS, a Retrospective Observational Study of Trifluridine/Tipiracil in Chemorefractory Metastatic Colorectal Cancer"

_biomedicines, 2023, doi:10.3390/biomedicines11051267_

Round 1
Reviewer 1 Report
Journal: Biomedicines
Title: RETRO-TAS, a Retrospective Observational Study of Tri-fluridine/Tipiracil in Chemorefractory Metastatic Colorectal Cancer
The authors: Anna Koumarianou, et al.
This retrospective analysis (The RETRO-TAS study) provides real world population-based data on FTD/TPI treatment in patients with mCRC. Median OS and PFS from initiation of FTD/TPI in the total cohort were 11.4 and 4.8 months, respectively. Those data of OS and PFS are superior compared to previous reports. The authors attributed the favorable treatment outcome to a higher proportion of patients being treated in the 3rd line compared to previous reports.
This study provides real-world data for FTD/TPI treatment in patients with mCRC and is very informative for physicians engaged in chemotherapy for mCRC.
Comments and questions
The authors showed the mutations status of KRAS (52%), NRAS (5%), HER2 (3.5%), BRAF (3.5%) and MSI (9%). Does mutation status influence therapeutic efficacy of FTD/TPI? Are there any cases in the MSI-H population who received immunotherapy such as anti PD-1 antibody? What is the effect on MSI-H patients?
Line 163: The prefix “c)” may be missing in the sentence.
Author Response
We thank the reviewer for the valuable time and effort towards the improvement of our manuscript. We appreciate it.
In relation to the mutational status, we found no relation with response to treatment. We have included these in the univariable model but it didn't surface as significant. In order to better indicate this finding we have now included an improved description in the last paragraph of "Efficacy" in the "Result" section (line 278-290).
In relation to the two very interesting questions on the MSI-H population, we did not include such subanalysis to study the effect of immunotherapy during the design of the study and we can not provide any data in this direction.
The prefix c) was added in line 163 as indicated
Reviewer 2 Report
Koumarianou et al. has made the real-world observational study confirms and adds on the findings of the RECOURSE phase III study in relation to the toxicity and the effectiveness of FTD/TPI in all subgroups of patients with chemotherapy refractory mCRC, regardless of mutational status and sidedness. The manuscript is well-written and the effectiveness and toxicity of TAS-102 has been addressed and I have no major criticisms. However, I would like to suggest the citation of the many related articles from the Asian fellow researchers for the comparison with previous chemotherapeutic agents with similar pharmaceutical mechanisms and predicting biomarkers:
1. Asian Journal of Surgery. Prognosis of unresectable stage IV Colon cancer with primary tumour resection. A multicenter study of minimally or asymptomatic primary tumour. Available online 14 December 2022
2. Asian Journal of Surgery. Volume 45, Issue 1, January 2022, Pages 448-455
3. Asian Journal of Surgery. Volume 44, Issue 5, May 2021, Pages 715-722
Author Response
We thank the reviewer for the valuable time and effort towards the improvement of our manuscript. We have carefully evaluated the comment on the manuscripts from China. We have already included two extremely significant publications on TAS-102 from China (references 7 and 24). The two references, indicated by the reviewer as numbers 2+3, from the "Asian Journal of Surgery" although extremely valuable in the field of colorectal cancer can not be accommodated in this manuscript as they are reporting on data relating to other drugs and in the adjuvant setting. The third reference indicated by the reviewer as number 1 from the "Asian Journal of Surgery" reports on the role of surgery in the metastatic setting. Reporting on these three references would move the focus of our manuscript towards other scientific pathways and outside of the scopus of this pharmacological study reporting on the efficacy of TAS-102 in the third line therapeutic setting of stage IV inoperable colorectal cancer.